# Serology in COVID-19: Comparison of Two Methods [note 1]

**DOI:** 10.3390/ijerph18126497

**Published:** 2021-06-16

**Authors:** Anna Moniuszko-Malinowska, Wojciech Jelski, Justyna Dunaj, Barbara Mroczko, Piotr Czupryna, Ewelina Kruszewska, Sławomir Pancewicz

**Affiliations:** 1Department of Infectious Diseases, Neuroinfections Medical University in Bialystok, Żurawia 14, 15-540 Białystok, Poland; dunaj.justyna@wp.pl (J.D.); avalon-5@wp.pl (P.C.); kruszewska.ewelina@gmail.com (E.K.); spancewicz@interia.pl (S.P.); 2Department of Biochemical Diagnostics, Medical University in Bialystok, Jerzego Waszyngtona 15A, 15-269 Białystok, Poland; wjelski@umb.edu.pl (W.J.); mroczko@umb.edu.pl (B.M.); 3Department of Neurodegeneration Diagnostics, Medical University in Bialystok, Jerzego Waszyngtona 15A, 15-269 Białystok, Poland

**Keywords:** SARS-CoV-2, chemiluminescent immunoassay, serology, spike protein, nucleocapsid protein

## Abstract

Background: The aim of our study was to examine the performance of two assays in detecting SARS-CoV-2 antibodies. Methods: A total of 127 COVID-19 disease contacts from the Infectious Diseases Department were included. Two serological tests were used: SARS-CoV-2 IgG CMIA on the Alinity system (Abbott) and LIAISON^®^ SARS-CoV-2 S1/S2 IgG CLIA (DiaSorin). Results: The assays exhibited a 96.85% (123/127 patients) test result agreement. In two cases, the positive results obtained by SARS-CoV-2 IgG CMIA on the Alinity system (Abbott) were negative based on the LIAISON^®^ SARS-CoV-2 S1/S2 IgG CLIA (DiaSorin) test, and in two cases, negative results from the LIAISON^®^ SARS-CoV-2 S1/S2 IgG CLIA (DiaSorin) test were positive with the SARS-CoV-2 IgG CMIA on the Alinity system (Abbott). Conclusions: Based on the results of our study, we conclude that in population medicine, the assessments of anti-SARS-CoV-2 antibodies after exposure to SARS-CoV-2 virus based on spike protein or nucleocapsid protein show comparable effectiveness.

## 1. Introduction

Validated serologic assays are crucial for epidemiological investigation and identification of viral reservoir hosts. Epidemiological studies are urgently needed to help uncover the burden of disease, particularly the rate of asymptomatic infections, and to better estimate morbidity and mortality. Additionally, these epidemiological studies can help reveal the extent of viral spread in households, communities, and specific settings, which could help guide control measures.

An important application of serological tests is to understand the antibody responses associated with SARS-CoV-2 infection and vaccination. Another important role of serological studies is to provide answers concerning potential reinfection. It is unknown whether the presence of binding antibody to the spike (S) protein or the receptor-binding domain (RBD) antigens correlates with virus neutralization and whether antibody titers (binding or neutralizing) correlate with protection from reinfection. Serology testing is also useful in the evaluation of antibody titers in donors for convalescent plasma therapy [1,2,3].

More and more commercial tests are becoming available, and evaluation of their usefulness is needed. We aimed to compare the usefulness of two different immunoassays in anti-SARS-CoV-2 antibody detection (against N protein and against S protein) because of the potential diversity of coronavirus strains and the variety of immunological responses after infection in the population.

## 2. Materials and Methods

### 2.1. Materials

In total, 127 patients exposed to SARS-CoV-2 were included in the study. Among them, 68 (53.54%) were patients hospitalized at the Department of Infectious Diseases and Neuroinfections at the Medical University of Bialystok, Poland, and 59 (46.46%) were healthcare workers who were exposed to the virus but did not have symptoms of infection. The healthcare worker group consisted of personnel who worked with COVID-19 patients in the infectious diseases ward since the beginning of the pandemic. They were in close contact with SARS-CoV-2-positive patients but always wore PPE.

The diagnosis of SARS-CoV-2 infection was confirmed by reverse transcription-polymerase chain reaction (RT-PCR) testing using the CFX96 Real-Time System (Bio-Rad) from nasopharyngeal swabs.

Blood samples for immunoserological diagnosis were collected from all patients in the study one month after exposure.

The study was approved by the Bioethical Commission of Medical University of Bialystok (APK.002.259.2020).

### 2.2. Methods

Two serological tests were used: The SARS-CoV-2 IgG chemiluminescent microparticle immunoassay (CMIA) used for the qualitative detection of anti-N protein IgG antibodies to SARS-CoV-2 in human plasma or serum on the Alinity system (Abbott)The LIAISON^®^ SARS-CoV-2 S1/S2 IgG chemiluminescence immunoassay (CLIA) technology for the quantitative determination of anti-S1 and anti-S2 specific IgG antibodies to SARS-CoV-2 (DiaSorin)

#### 2.2.1. SARS-CoV-2 IgG—Alinity (Abbott)

The SARS-CoV-2 IgG assay is a chemiluminescent microparticle immunoassay (CMIA) used for the qualitative detection of IgG antibodies to SARS-CoV-2 in human plasma or serum on the Alinity system (Abbott). This assay is an automated two-step immunoassay. The IgG antibodies to SARS-CoV-2 present in the sample bound to the SARS-CoV-2 antigen-coated microparticles. Anti-human IgG acridinium-labelled conjugate was added to create a reaction mixture, which was incubated. The resulting chemiluminescent reaction was measured as a relative light unit (RLU). There is a direct relationship between the number of IgG antibodies to SARS-CoV-2 in the sample and the RLU detected by the system optics. This relationship is reflected in the calculated index (S/C). A titer above 1.4 was considered positive.

#### 2.2.2. LIAISON^®^ SARS-CoV-2 S1/S2 IgG

The specific recombinant S1 and S2 antigens were used for coating magnetic particles (solid phase), and mouse monoclonal antibodies to human IgG were linked to an isoluminol derivative (isoluminol–antibody conjugate). During the first incubation, the SARS-CoV-2 IgG antibodies present in calibrators, samples, or controls bound to the solid phase through the recombinant S1 and S2 antigens. During the second incubation, the antibody conjugate reacted with the IgG to SARS-CoV-2 already bound to the solid phase. After each incubation, the unbound material was removed with a wash cycle. Subsequently, the starter reagents were added, and a flash chemiluminescence reaction was thus induced. The light signal, and hence the amount of isoluminol-antibody conjugate, was measured by a photomultiplier as relative light units (RLU) and was indicative of IgG to SARS-CoV-2 concentration present in calibrators, samples, or controls. A titer above 15 AU/mL was considered positive.

### 2.3. Statistical Analysis

Statistical analysis was performed by using STATISTICA Data Miner + QC. For immeasurable features, percentages were calculated. A *p*-value < 0.05 was considered statistically significant.

## 3. Results

The anti-SARS-CoV-2 IgG antibodies matched when detecting N and S protein in 123 of 127 (96.85%) patients. The results differed in only 4 of 127 cases. In two cases, the positive results in the test based on N were negative in the test based on S, and in two cases, the negative results in the test based on S were positive in the test based on N (*p* = NS). All four patients had a mild course of the disease and recovered completely (Table 1).

In 56 of 127 (44.09%) patients, anti-SARS-CoV-2 IgG antibodies were detected; 55 of these 56 patients (98.2%) were symptomatic. None of the healthcare workers were positive. The mean titer of anti-SARS-CoV-2 IgG antibodies when detecting N protein was 2.75 ± 3.34 (max: 9.75 S/C), while the mean titer of anti-SARS-CoV-2 IgG antibodies when detecting S protein was 90.64 ± 68.97 (max: 330 AU/Ml).

## 4. Discussion

There are many different assays for the detection of anti-SARS-CoV-2 antibodies. Assays are designed to detect immunoglobulin class G (IgG) antibodies to the nucleocapsid protein (protein N) or spike protein (the main surface glycoprotein that the virus uses to attach and enter cells) of SARS-CoV-2. Other assays detect RBD, which is part of the spike protein (anti-S1 and anti-S2). They are designed to be performed using the serum and plasma from patients with signs and symptoms of infection who are suspected to have coronavirus disease or patients that may have been infected by SARS-CoV-2.

The spike and nucleocapsid proteins are major immunogenic components of CoVs and are produced in abundant quantities during infection. The S protein is the primary determinant of protective immunity and cross-species transmission in CoVs, and monoclonal antibodies against the S protein could neutralize viral infectivity. On this basis, Walls et al. hypothesized that exposure to either SARS-CoV or SARS-CoV-2 would elicit a mutually cross-reactive response, potentially neutralizing antibodies, and demonstrated the ability of plasma from four mice immunized with a SARS-CoV S protein to bind SARS-CoV-2 S protein and block SARS-CoV-2 entry into target cells [4].

In clinical practice, the role of serological tests is extremely valuable in population medicine for patients who previously had COVID-19 and plan to be blood donors. The sensitivity and specificity of these tests should be as high as possible.

Our study shows that both tests have comparable values in the assessment of immunity after infection, as the assays exhibited 96.85% overall compatibility. The results of our study are in opposition to a study by Burbelo et al., who concluded that antibodies against the nucleocapsid protein of SARS-CoV-2 were more sensitive than spike protein antibodies for detecting early infection [5]. In addition, Charlton et al. observed that CMIA detecting IgG antibodies against recombined nucleocapsid protein (Abbott) had a higher sensitivity than the anti-spike protein antibody detection assay (DiaSorin) in both negative and positive samples [6]. On the other hand, Honemann et al. observed that assay performance was independent of the usage of either nucleocapsid or spike proteins [7]. Cerino et al. reported good agreement for Abbott (Cohen’s kappa coefficient: 0.69) and moderate agreement for Liaison (Cohen’s kappa coefficient: 0.58); however, their study showed that Abbott and Liaison SARS-CoV-2 CLIA IgG had good agreement in seroprevalence assessment [8]. Although tests based on antibodies against N protein seem to be more sensitive than those based on anti-S antibodies, it should be emphasized that S-based tests are more specific because of the lower probability of cross-reactivity. Moreover, the immune response against S antigen seems to appear earlier compared with the response to *N* antigen (Cerino) [2,9].

Our study shows the usefulness of both tests, especially for epidemiological purposes. In the majority of symptomatic patients, infection with SARS-CoV-2 leads to immunization against both proteins; therefore, it is important to detect antibodies with high quality kits.

A limitation of our study was the small sample size, so further studies are needed.

## 5. Conclusions

The assessment of anti-SARS-CoV-2 antibodies in population medicine based on spike protein or nucleocapsid protein shows comparable usefulness.

## Figures and Tables

**Table 1 ijerph-18-06497-t001:** Characteristics of four patients with conflicting results of immunological tests.

Patients No.	Sex	Age	COVID-19 Severity	Comorbidities	Anti-N Antibodies	Anti-S Antibodies	WBC at Admission	Lymphocytes at Admission	Neutrophiles at Admission	Recovery
I	M	22	mild	none	negative	30.5	4110	1450	1920	complete
II	M	45	mild	none	negative	18.2	3860	550	2610	complete
III	F	34	mild	none	2.74	negative	3530	1060	2110	complete
IV	F	46	mild	none	3.15	negative	4990	860	3440	complete

## Data Availability

The data that support the findings will be available on request under the corresponding author’s e-mail: annamoniuszko@op.pl.

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
