# Peer review of "Serology in COVID-19: Comparison of Two Methodsâ€"

_ijerph, 2021, doi:10.3390/ijerph18126497_

Round 1
Reviewer 1 Report
This brief report compares the performance of the two assays in detecting SARS-CoV-2 antibodies, SARS-CoV-2 anti-N IgG CMIA and anti-SARS-CoV-2 S1/S2 IgG. Tested assays exhibited 96.85% % overall compatibility, and the authors conclude that both of them are suitable for evaluation of immunity after SARS-CoV-2 infection.
The study is well-designed, the sample size is sufficient to evaluate the tests and the conclusions made by the authors are supported by data. However there are some points that would benefit from further clarification:
- it is unclear what the authors define as virus exposure. They describe the healthcare workers as asymptomatic, but it is unclear to me if these people ever tested positive for SARS-CoV-2, were included based on their close contact with SARS-CoV-2 positive patients but were still protected from infection by PPE, or perhaps were accidently exposed to the virus in some way. This should be clarified. I have similar questions about the included patients, were all of them COVID-19 patients, or were some of them hospitalized for other reasons? Did they all test positive for SARS-CoV-2 at the time of infection?
- it would be worth clarifying if the samples that came up positive in only one out of the two tests were strongly positive, or borderline positive, and whether weakly positive results could perhaps contribute to these rare discrepancies in test performance
- There are a few typos in the manuscript, for example "orotein" in line 37 and "men titre" in lines 97 and 99.
Author Response
Dear Reviewer,
Thank You very much for Your comments regarding the manuscript titled: Serology in COVID-19 – comparison of two methods.
Please find our response below:
Rev.1
This brief report compares the performance of the two assays in detecting SARS-CoV-2 antibodies, SARS-CoV-2 anti-N IgG CMIA and anti-SARS-CoV-2 S1/S2 IgG. Tested assays exhibited 96.85% % overall compatibility, and the authors conclude that both of them are suitable for evaluation of immunity after SARS-CoV-2 infection.
The study is well-designed, the sample size is sufficient to evaluate the tests and the conclusions made by the authors are supported by data.
However there are some points that would benefit from further clarification:
- it is unclear what the authors define as virus exposure. They describe the healthcare workers as asymptomatic, but it is unclear to me if these people ever tested positive for SARS-CoV-2, were included based on their close contact with SARS-CoV-2 positive patients but were still protected from infection by PPE, or perhaps were accidently exposed to the virus in some way. This should be clarified. I have similar questions about the included patients, were all of them COVID-19 patients, or were some of them hospitalized for other reasons? Did they all test positive for SARS-CoV-2 at the time of infection?
We thank Reviewer for this remark. We added the necessary information to manuscript.
The healthcare workers group consisted of personnel working with COVID-19 patients since the beginning of pandemic in the infectious diseases ward. They were in the close contact with SARS-CoV-2 positive patients, but always wearing PPE.
Patients included to the study were all COVID-19 patients, hospitalized in the same infectious diseases ward. The diagnosis of SARS-CoV-2 infection was confirmed by reverse transcription-polymerase chain reaction (RT-PCR) testing by the CFX96 Real-Time System (Bio-Rad) from nasopharyngeal swabs.
- it would be worth clarifying if the samples that came up positive in only one out of the two tests were strongly positive, or borderline positive, and whether weakly positive results could perhaps contribute to these rare discrepancies in test performance
The conflicting results varied greatly and included: strongly positive vs strongly negative (2 patients) and strongly positive vs borderline (2 patients), therefore no specific trend could be distinguished.
These discrepancies may be due to other sites of protein attachment on immunoglobulin, fact that antigen presentation and trafficking vary between different epitopes, or the individual characteristics of the patient's immune system and the diversity of our society. Our study confirms, that used tests are close to product characteristics when considering the sensitivity and specifity of the immunoserological tests (Abbott: sensitivity – 100% after 14 days; specificity – 99.93%; DiaSorin: sensitivity – 97.4% after 15 days; specificity – 98.5%%.
Moreover, it is worth to remember, that in some cases, especially patients with immunodeficiency various tests may give various results.
- There are a few typos in the manuscript, for example "orotein" in line 37 and "men titre" in lines 97 and 99.
It was corrected.
Reviewer 2 Report
In this paper, the authors have compared 2 methods of detection of SARS-CoV-2 antibodies in 68 patients hospitalized for COVID-19 and in 59 healthcare workers. They conclude that both technics, based on different epitopes (spike or nucleocapsid) are comparable.
Majors comments:
- In the patient group, how was diagnosed SARS-CoV-2 infection ? It should be described. A better description of the patients is mandatory: blood was collected at least one month after the exposition, but the onset of symptoms should be indicated, as well as the average delay between diagnosis of COVID and antibodies research.
- Two patients, for each method, were negative despite SARS-CoV-2 infection. The authors should describe the characterictics of these fours patients, particularly concerning the severity of the disease, since it has been shown that the level of antibodies could be low in patients with a mild disease; It should give some clues which kind of assay (against N or S) is appropriate according to clinical data, for example.
- Important papers concerning the comparisons of serological assays are not indicated in the references (PMID: 33996728, PMID: 33930691). Bibliography should be more exhaustive (but perhaps these paper were published when this paper was submitted)
- The discussion should take into account the papers cited above concerning the evaluation of these assays and if possible, propose a strategy to use one of these tests according to the patient characteristics.
Minor comments:
-line 37 : typo "orotein" instead of "protein"; the abbreviation RBD is not define, but is defined in the discussion line 105)
- line 97 and 98: "men" instead of "mean"
Author Response
Dear Reviewer,
Thank You very much for Your comments regarding the manuscript titled: Serology in COVID-19 – comparison of two methods.
Please find our response below:
Rev.2
In this paper, the authors have compared 2 methods of detection of SARS-CoV-2 antibodies in 68 patients hospitalized for COVID-19 and in 59 healthcare workers. They conclude that both technics, based on different epitopes (spike or nucleocapsid) are comparable.
Majors comments:
In the patient group, how was diagnosed SARS-CoV-2 infection ? It should be described. A better description of the patients is mandatory: blood was collected at least one month after the exposition, but the onset of symptoms should be indicated, as well as the average delay between diagnosis of COVID and antibodies research.
We thank Reviewer for this remark. We added the necessary information to manuscript.
Patients included to the study were all COVID-19 patients, hospitalized in the infectious diseases ward, as healthcare workers. The diagnosis of SARS-CoV-2 infection was confirmed by reverse transcription-polymerase chain reaction (RT-PCR) testing by the CFX96 Real-Time System (Bio-Rad) from nasopharyngeal swabs. Knowing that serology assays can detect SARS-CoV-2 antibodies as early as 10 days after symptom onset, but the sensitivity increases with time from of the symptoms onset (significantly at least 15 days after the symptoms onset) we chose patients ca. one month (21 – 35 days) after the symptoms appearance.
Two patients, for each method, were negative despite SARS-CoV-2 infection. The authors should describe the characterictics of these fours patients, particularly concerning the severity of the disease, since it has been shown that the level of antibodies could be low in patients with a mild disease; It should give some clues which kind of assay (against N or S) is appropriate according to clinical data, for example.
We thank Reviewer for this remark. We added the necessary information in the table to manuscript. We noticed that all 4 patients has mild course of the disease and recover completely.
|
Patients No |
Sex |
Age |
COVID-19 severity |
Comorbidities |
Anti-N antibodies |
Anti-S antibodies |
WBC at admission |
Lymphocytes at admission |
Neutrophiles at admission |
Recovery |
|
I |
M |
22 |
mild |
none |
negative |
30.5 |
4110 |
1450 |
1920 |
complete |
|
II |
M |
45 |
mild |
none |
negative |
18.2 |
3860 |
550 |
2610 |
complete |
|
III |
F |
34 |
mild |
none |
2.74 |
negative |
3530 |
1060 |
2110 |
complete |
|
IV |
F |
46 |
mild |
none |
3.15 |
negative |
4990 |
860 |
3440 |
complete |
Important papers concerning the comparisons of serological assays are not indicated in the references (PMID: 33996728, PMID: 33930691). Bibliography should be more exhaustive (but perhaps these paper were published when this paper was submitted)
We thank Reviewer for this remark. It is true, that our manuscript was written few months ago and was being kept in one journal over 3 months, as other papers have been already published. We added the suggested references.
The discussion should take into account the papers cited above concerning the evaluation of these assays and if possible, propose a strategy to use one of these tests according to the patient characteristics.
We thank Reviewer for this remark. We extended the discussion using the suggested papers and two others. Moreover, we emphasized the usefulness of both tests in epidemiological studies, as we have not proven priority of none of two compared tests.
Minor comments:
-line 37 : typo "orotein" instead of "protein"; the abbreviation RBD is not define, but is defined in the discussion line 105)
It was corrected.
- line 97 and 98: "men" instead of "mean"
It was corrected.